

# Out of (the) bag—encoding categorical predictors impacts out-of-bag samples

Helen L. Smith[1], Patrick J. Biggs[2,3,4], Nigel P. French[3,4], Adam N. H. Smith[5] and Jonathan C. Marshall[1]

[1] School of Mathematical and Computational Sciences, Massey University, Palmerston North, New Zealand
[2] School of Food Technology and Natural Sciences, Massey University, Palmerston North, New Zealand
[3] NZ Food Safety and Science Research Centre, Massey University, Palmerston North, New Zealand
[4] School of Veterinary Science, Massey University, Palmerston North, New Zealand
[5] School of Mathematical and Computational Sciences, Massey University, Auckland, New Zealand

## ABSTRACT

Performance of random forest classification models is often assessed and interpreted using out-of-bag (OOB) samples. Observations which are OOB when a tree is trained may serve as a test set for that tree and predictions from the OOB observations used to calculate OOB error and variable importance measures (VIM). OOB errors are popular because they are fast to compute and, for large samples, are a good estimate of the true prediction error. In this study, we investigate how target-based *vs.* target-agnostic encoding of categorical predictor variables for random forest can bias performance measures based on OOB samples. We show that, when categorical variables are encoded using a target-based encoding method, and when the encoding takes place prior to bagging, the OOB sample can underestimate the true misclassification rate, and overestimate variable importance. We recommend using a separate test data set when evaluating variable importance and/or predictive performance of tree based methods that utilise a target-based encoding method.

## INTRODUCTION

### Out-of-Bag sample

Random forest classification is a method of supervised machine learning that creates an ensemble of classification trees. The individual trees that make up the ensemble differ from one another because they are each trained on a different random sample of predictor variables ('random subspacing'; *Amit & Geman, 1997*; *Breiman, 1996*; *Ho, 1998*). In addition, each individual tree is trained on a different bootstrap sample of the observations in the training set ('bagging' or 'bootstrap aggregating'). The bootstrap sample for each tree ("the bag") typically contains about two-thirds of the observations in the training data. The remaining one-third of observations are "out-of-bag" (OOB) and serve as a test set for the tree. The OOB sample may be used to estimate the predictive performance of the random forest and variable importance measures (VIM), amongst other things.

Corresponding author
Helen L. Smith,
h.l.smith@massey.ac.nz

## Out-of-Bag error

An OOB prediction for an observation is obtained by aggregating the tree classifications for the observation from the OOB samples. The misclassification rate of the OOB predictions from all training observations is the OOB error (*Breiman, 2001*) (Fig. 1). OOB errors are popular because they are fast to compute, requiring only a single random forest to be computed, and have been reported to be a good estimate of the true prediction error (*Adelabu, Mutanga & Adam, 2015*; *Lawrence, Wood & Sheley, 2006*; *Mutanga & Adam, 2011*). The OOB error may also be used to select appropriate values for tuning parameters, such as the number of predictor variables that are randomly drawn for a split[1]. *Breiman (1996, 2001)* claimed that the OOB error alleviates the need for cross-validation or setting aside a separate test set; however, it has been shown that, especially for small samples, the OOB error can over-estimate the true prediction error (*Bylander, 2002*; *Mitchell, 2011*; *Janitza & Hornung, 2018*). Methods to address the bias have been proposed (*Bylander, 2002*; *Mitchell, 2011*; *Janitza & Hornung, 2018*), although, when available, a large external validation data set will provide a more precise error estimate, serving as a gold standard (*Hastie, Tibshirani & Friedman, 2009*; *Janitza & Hornung, 2018*).

## Variable importance

OOB samples may also be used to calculate measures of variable importance. VIM can be used to rank predictor variables according to their degree of influence on the predicted outcomes. There are two broad measures of variable importance for random forests—the Mean Decrease in Accuracy (MDA, or permutation importance) (*Breiman, 2001*); and the Mean Decrease in Impurity (MDI, or Gini importance) (*Breiman, 2002*). For both measures, a high value means that the variable has a positive impact on predictions.

MDA for a given variable is the mean decrease in prediction accuracy of the individual trees across the forest when the variable is not used for prediction. MDA is obtained by permuting values of the variable in the OOB sample and computing the difference in the error rate on the permuted OOB sample from the original OOB sample (Fig. 2). The idea is that permuting an important variable would result in a large decrease in accuracy while permuting an unimportant variable would have a negligible effect.

MDI is the weighted mean of the individual trees' decrease of impurity produced by a given variable. An important variable is expected to generate a larger decrease of impurity (*i.e.* more pure splits) than an unimportant variable. The decrease of impurity is measured as the difference between a node's Gini impurity and the weighted sum of the Gini impurity of the two child nodes, evaluated on the in-bag samples.

Several studies have highlighted issues with these importance measures and have proposed modifications which may overcome specific undesirable properties (*Strobl et al., 2007, 2008*; *Sandri & Zuccolotto, 2008*; *Nicodemus & Malley, 2009*; *Nicodemus, 2011*; *Toloşi & Lengauer, 2011*; *Janitza, Celik & Boulesteix, 2018*; *Gregorutti, Michel & Saint-Pierre, 2017*; *Nembrini, König & Wright, 2018*; *Benard, Sebastien & Scornet, 2022*; *Mentch & Zhou, 2022*; *Wallace et al., 2023*; *Williamson et al., 2023*). *Janitza, Celik & Boulesteix (2018)* introduced the Holdout variable importance method which computes MDA using a

[1] Referred to as `mtry` in R packages `ranger`, `randomForest`, `randomForestSRC`, and the tidymodels framework; or `max_features` in Python's sklearn `RandomForestClassifier`.

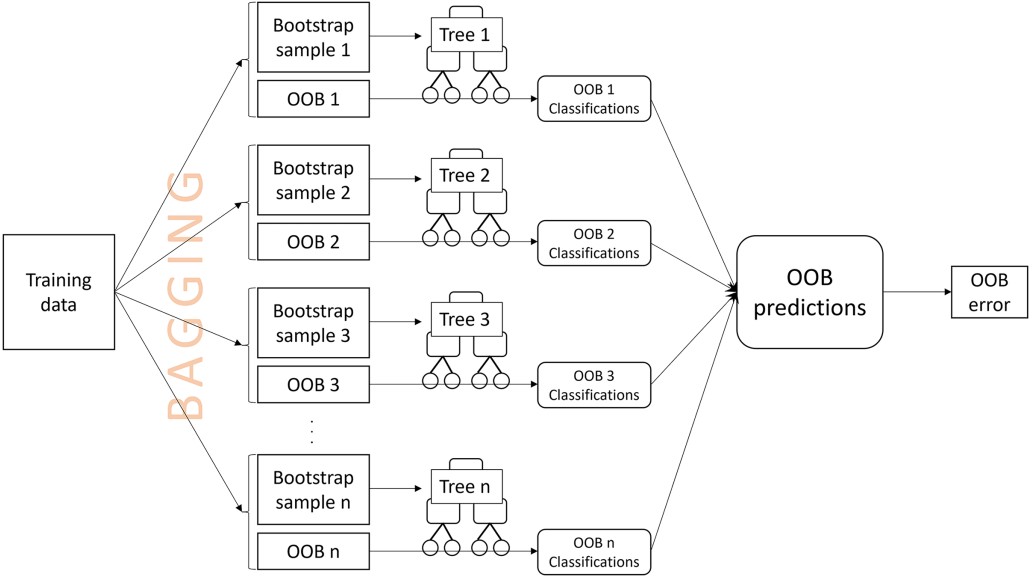

**Figure 1 A visual description of the process of obtaining an out-of-bag (OOB) error estimate.**

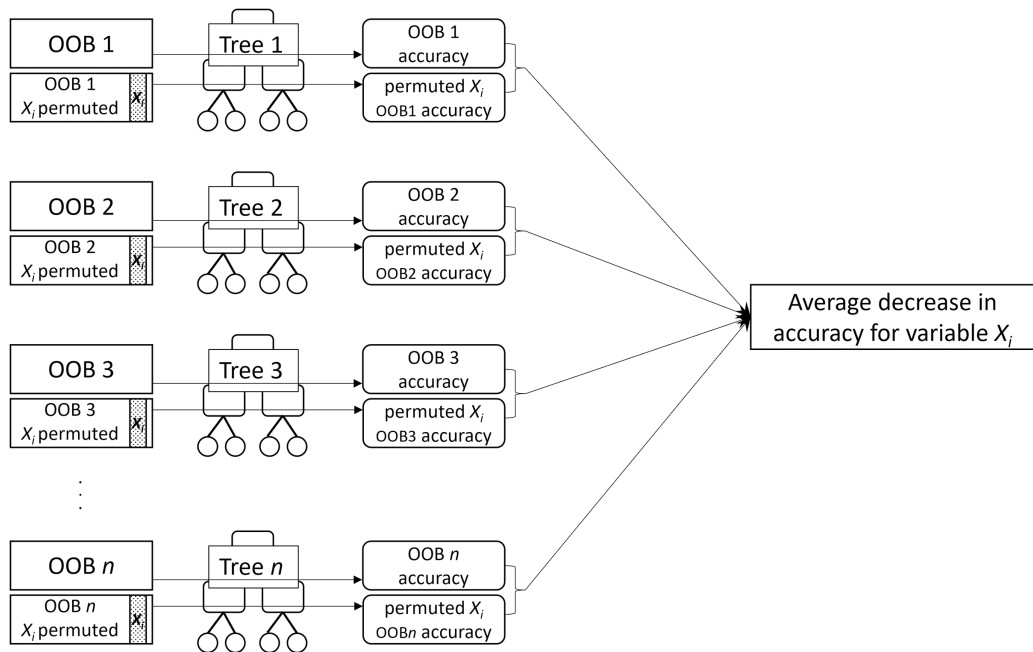

**Figure 2 A visual description of the process of obtaining permutation importance (MDA) for variable $X_i$.**

second cross-validation fold rather than the OOB data and has been adopted as an option by the `ranger` and `randomForestSRC` packages. Also implemented by `ranger` is the actual impurity reduction (AIR) importance method (*Sandri & Zuccolotto, 2008*; *Nembrini, König & Wright, 2018*) which adjusts the original impurity by subtracting the

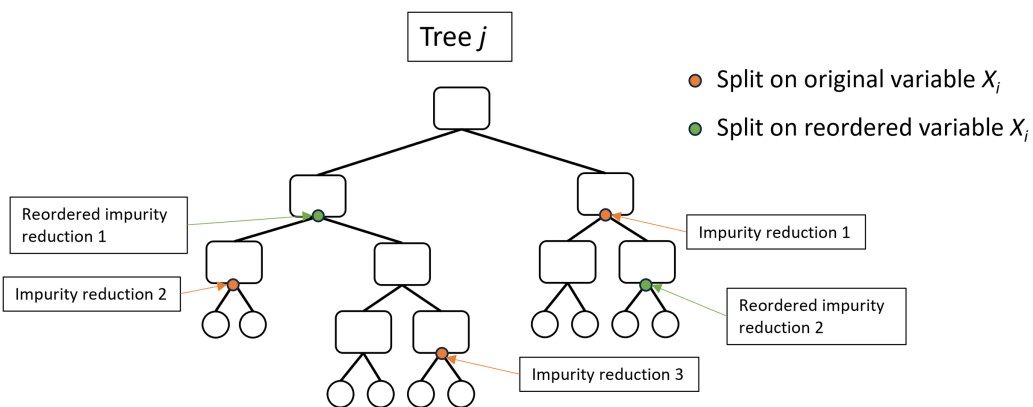

**Figure 3 Illustration of the actual impurity reduction (AIR) calculation.** The AIR for variable $X_i$ for $\text{Tree}_j = \sum \text{Impurity reduction}_{X_i} - \sum \text{Impurity reduction}_{reorderedX_i}$, where impurity reduction is the Gini impurity of the parent node minus the weighted sum of the Gini impurity of the two child nodes.

impurity importance following random reordering of the variable (Fig. 3). There have been many other variable importance measures proposed (*e.g. Loecher (2022)*, *Epifanio (2017)*, *Dfuf et al. (2020)*), however they have not been widely adopted and MDA is generally considered the most efficient and accurate measure of variable importance (*Ishwaran, 2007*; *Strobl et al., 2007*; *Nicodemus et al., 2010*; *Boulesteix et al., 2012*; *Ziegler & König, 2014*; *Szymczak et al., 2016*).

## Encoding categorical predictors

Categorical variables can, in theory, be used by random forests in their raw state; however in practice, software will either require them to be numerically encoded (*Pedregosa et al., 2011*) or will encode them prior to processing (*Wright & Ziegler, 2017*; *Liaw & Wiener, 2002*). There are several methods of encoding categorical variables. Ordinal encoding of categorical predictors has several benefits, including increased computational efficiency, evading restrictions on the number of predictor categories[2], and managing absent levels (*Au, 2018*; *Smith et al., 2024*). The encoding method can be independent of the response variable (*i.e.* target-agnostic methods, such as one-hot encoding, integer encoding, and PCO-encoding (*Smith et al., 2024*)) or can incorporate information about the target values associated with a given level (*i.e.* target-based methods, such as CA-encoding (*Coppersmith, Hong & Hosking, 1999*; *Wright & Ziegler, 2017*) and CA-unbiased-encoding (*Smith et al., 2024*)).

Encoding may be performed at different stages of the algorithm (Fig. 4). The most computationally efficient method is to encode the predictor variables prior to bagging (*i.e.*, once on the entire dataset rather than each sub-sample undergoing encoding independently) (*Wright & Ziegler, 2017*). Encoding can also take place after bagging (*i.e.*, on each sub-sample or at each split in the tree (*Breiman, 1996*; *Liaw & Wiener, 2002*)); however, this has a much higher computational cost.

Target-based encoding methods necessarily have information leakage from the target variable to the predictors. If a predictor is encoded prior to splitting into training and test

[2] When nominal encoding a categorical variable, each binary node assignment is saved using the bit representation of a double integer, which limits this treatment to predictors with fewer than 54 levels (*Wright & König, 2019*).

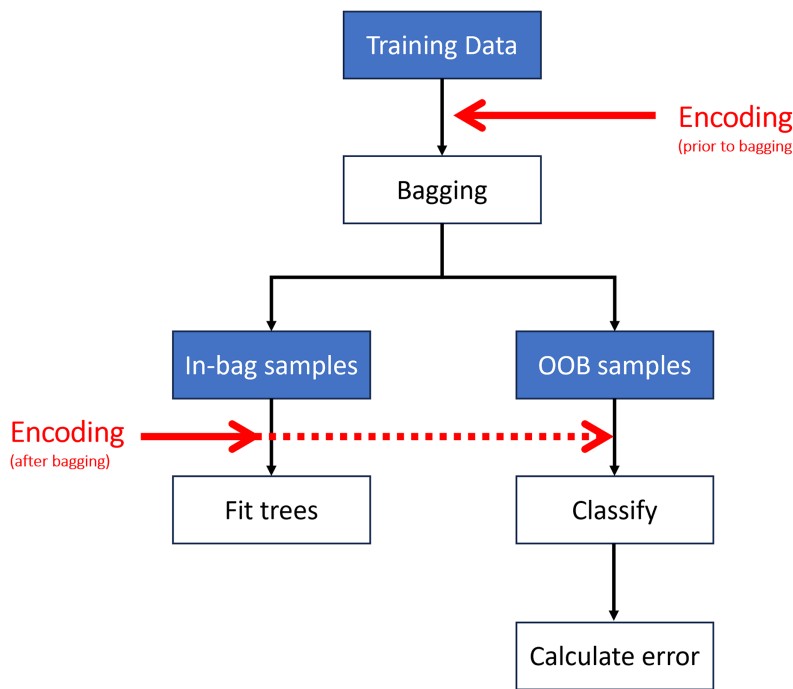

**Figure 4  Encoding may take place prior to or after creating the out-of-bag (OOB) samples.**

sets, information from the target variable in the test set will leak to the predictors in the training set by way of the *a priori* encoding. In the same way, if a predictor is encoded prior to bagging, information from the target variable in the bootstrap samples will leak to the predictors in the OOB samples. The OOB observations will not, therefore, behave like fully independent test data. Target-agnostic encoding methods do not have this issue with information-leakage because the response class (target) information is not used for the encoding.

Treating the OOB samples like an independent test set is therefore only reasonable if a target-agnostic encoding method is used, or if a target-based encoding method is performed subsequent to bagging. Otherwise, calculating misclassification rates and measures of variable importance on the OOB sample, or indeed the encoded variables, as in the case of the Holdout variable importance (*Janitza, Celik & Boulesteix, 2018*; *Wright & Ziegler, 2017*), is likely to underestimate the true error rate and overestimate the variable importance. The impact of method and timing of encoding has not been explicitly examined with regards to random forest OOB sample calculations.

## Study aims and objectives

Encoding of categorical variables is a necessary preprocessing step for many machine learning algorithms. The computational benefits of ordinal encoding categorical variables are well known. To our knowledge, the potential leakage of target information to the OOB samples as a result of target encoding categorical variables prior to bagging is unreported. Current debates lie in the accuracy of OOB error estimates and/or VIMs, particularly for

small sample sizes and unbalanced designs, but there appears to be no awareness that OOB samples may not be as 'good as an independent test set' and it remains a commonly held belief that OOB samples replace the need for separate test data.

For some popular random forest implementations (*e.g.* the R package `ranger` (*Wright & Ziegler, 2017*)), target encoding of categorical predictors prior to bagging is the recommended approach (*Wright & König, 2019*), and is performed internally within the method, in parallel with OOB error and VIM calculations. This has potentially resulted in biased and even misleading results in a number of studies.

In this study, we investigate the accuracy of OOB error estimates and variable importance measures when nominal categorical variables are ordinal encoded prior to bagging in random forest models. We compare how target-based *vs.* target-agnostic encoding of categorical predictor variables can affect the OOB error and estimates of variable importance using a random noise simulation study. We demonstrate that when target-based encoding is performed prior to bagging, OOB samples are biased due to information leakage from the target variable during the encoding process and we recommend using a separate test set instead of the OOB sample, or else to perform the encoding after bagging.

Although here we focus on random forest which incorporates bagging as a key component of the method, these results are generalisable to any applications which employ bagging (bootstrap aggregating), including other ensemble learning techniques; classification and regression tasks (*Dfuf et al., 2020*); outlier predictions (*Mohandoss, Shi & Suo, 2021*); feature selection (*Deviaene et al., 2019*; *Calle et al., 2011*; *Díaz-Uriarte & Alvarez de Andrés, 2006*); model tuning (*Adesina, 2022*); Gini-OOB index (*Chen, Tan & Yang, 2023*); and clustering (*Schumacher et al., 2016*; *Bigdeli, Maghsoudi & Ghezelbash, 2022*).

The aim of this study is to raise awareness of this simple, yet important and previously unreported, issue. Specifically, our goals are to:

(i) demonstrate why the common practice of using OOB samples instead of independent test data can lead to biased and potentially misleading results due to information leakage from the target variable during the process of encoding categorical predictors;

(ii) investigate *via* a short simulation study the accuracy of OOB error estimates and variable importance measures when nominal categorical variables are ordinal encoded prior to bagging in random forest models;

(iii) highlight the benefits of using independent test data for calculation of error estimates and variable importance measures; and

(iv) introduce the new 'Independent Holdout method' for calculating variable importance.

This article is structured as follows: "Introduction" includes a concise literature review and highlights the need for this research; "Methods" discusses implementation specific treatment of categorical variables, and describes the simulation methodology including data generation; "Results" presents the results of a short simulation study on OOB error and VIM measurements; "Discussion" answers the research questions and discusses the implications of our results; "Conclusion" summarises the findings of the study.

**Table 1 Implementation specific treatment of categorical variables.**

| Implementation | Predictor type | Predictor treatment | Handles absent levels | Timing of encoding | Maximum levels |
|---|---|---|---|---|---|
| ranger | character vector | converts to unordered factor | yes | – | – |
| | ordered factor | treats as ordinal | yes | – | – |
| | unordered factor | exhaustive partition | yes | – | 53 levels |
| | | orders alphabetically | yes | before bagging | – |
| | | target encodes | yes | before bagging | – |
| randomForest | character vector | orders alphabetically | yes[1] | before bagging | – |
| | ordered factor | treats as ordinal | yes[1] | – | 53 levels |
| | unordered factor | exhaustive partition | no | – | 53 levels |
| | | target encodes[2] | no | after bagging | 53 levels |
| randomForestSRC | character vector | unable to process | – | – | – |
| | ordered factor | treats as ordinal | yes[3] | – | – |
| | unordered factor | partial partition | yes[3] | – | – |
| scikit-learn | character vector | one hot encoding | yes[3] | before bagging | – |
| | ordered factor | one hot encoding | yes[3] | before bagging | – |
| | unordered factor | one hot encoding | yes[3] | before bagging | – |

**Notes:**
[1] The absent levels need to be ordered last for consistency of encoding with the training set.
[2] Optimisation is employed in the case of 2-class classification when there are more than ten levels of a predictor variable.
[3] Treats absent levels as missing values.

## METHODS

### Implementation

There are many popular implementations of random forest, including over 20 packages in R (https://koalaverse.github.io/machine-learning-in-R/random-forest.html#random-forest-software-in-r) as well as the widely used Python machine learning library scikit-learn (*Pedregosa et al., 2011*). There is no single best implementation and most are optimised for some special property of the data (*Wright & Ziegler, 2017*). Algorithms do, however, differ in their treatment of categorical variables, including absent levels (*i.e.* levels of a predictor variable that are present in data for prediction that were not present when the random forest was trained) (Table 1), which may impact predictions (*Au, 2018*; *Smith et al., 2024*) and performance measures calculated from OOB samples.

An unordered (nominal) categorical predictor with $k$ levels has $2^{k-1} - 1$ possible binary splits. A random forest algorithm may search the set of possible splits, either exhaustively (*e.g.* randomForest[3] (*Liaw & Wiener, 2002*) and ranger[4] (*Wright & König, 2019*)), or partially (*e.g.* randomForestSRC (https://www.randomforestsrc.org/articles/getstarted.html#allowable-data-types-and-factors) (*Ishwaran & Kogalur, 2023*)). As each binary node assignment is saved using the bit representation of a double integer the exhaustive search option is limited to predictors with fewer than 54 levels. If the categorical predictor is defined as a character vector (*i.e.* rather than an unordered factor) it may, by default, be encoded alphabetically (*e.g.* randomForest) rather than converted to a factor (*e.g.* ranger). This is problematic if a separate data set (*i.e.* for prediction) has a different set of

[3] This is the default option for randomForest in the case of multi-class classification or two-class classification with predictors which have fewer than 10 levels.

[4] When the argument respect.unordered.factors is set to "partition".

levels to those in the training set, in which case the ordinal encoding of the two sets will not match. This will occur if the observations for prediction contain only a subset of the levels from the training set, or if there are absent levels.

An ordered categorical predictor with $k$ levels can be treated the same way as a numerical predictor with $k$ unique ordered values and, at most, $k-1$ possible split points. Again, care needs to be taken when the levels in the data to be predicted do not match exactly the levels in the training set as, for some algorithms (*e.g.* `randomForest`), the encoding of the levels may not match. For the case of two-class classification, a nominal predictor variable with $k$ levels may be ordered by the proportion of observations with the second response class in each level. The ordering may occur at each split (*e.g.* `randomForest`[5]), or once prior to growing the forest (*e.g.* `ranger`[6]). Subsequently, treating these variables as ordinal leads to identical splits in the random forest optimisation as considering all possible 2-partitions of the $k$ predictor levels (*Breiman et al., 1984*; *Ripley, 1996*). For multi-class classifications, an order may be imposed on a nominal variable alphabetically (*e.g.* `ranger`[7]), or according to the first principal component of the weighted covariance matrix of class probabilities, following *Coppersmith, Hong & Hosking (1999)*[8] (*e.g.* `ranger`[9]). Ordering the variables once on the entire dataset prior to bagging, rather than at each split, is computationally efficient and negates the upper limit on the number of variable levels (*Wright & König, 2019*).

Some implementations of random forest require categorical variables to be one-hot encoded prior to analysis (*e.g.* Python's `scikit-learn`). This means a single predictor with $k$ levels is replaced by $k-1$ indicator variables. Now there will be only a single possible split point at each node but from $k-1$ indicator variables. Using this method, some of the category levels will be randomly ignored for each split, and so the original predictor will be represented by $j$ binary predictors, where $j \leq k-1$.

Treatment of absent levels also differs between implementations. Some algorithms are unable to process absent levels of unordered factors at all (*e.g.* `randomForest`). Some treat absent levels as missing values, or if there are no true missing values will map them to the child node that has the most samples (*e.g.* `scikit-lear` (https://scikit-learn.org/stable/modules/ensemble.html#random-forests) and `randomForestSRC`). And some will send all observations with an absent level to a particular branch at any given node (*e.g.* `ranger` (https://github.com/imbs-hl/ranger/blob/master/R/predict.R#L167)) (*Smith et al., 2024*).

The method of treatment of categorical variables, including absent levels, by four popular implementations of random forest is summarised in Table 1.

## Simulation study

To investigate the accuracy of internally calculated misclassification rates and variable importance under null conditions, a set of data was simulated and analysed with random forest.

The simulated data consisted of $n$ individuals, each with one predictor variable allocated uniformly and with replacement from $k$ levels. One of three classification labels were randomly assigned to each individual. There is no relationship between the response and the predictors. A subset containing 80% of the observations was used for training the

[5] This optimisation proceeds when the predictor variable has more than 10 unordered levels.

[6] When the argument `respect.unordered.factors` is set to "order" or TRUE.

[7] When the argument `respect.unordered.factors` is set to "ignore" or FALSE.

[8] *Coppersmith, Hong & Hosking (1999)* use the first principal component of the weighted matrix of class probabilities.

[9] When the argument `respect.unordered.factors` is set to "order" or TRUE.

random forest, and the remaining 20% of observations were used as the set of testing data. The process was repeated for each combination of sample size $n \in \{20, 50, 100, 150, 200, 400\}$ and number of variable levels $k \in \{1, 5, 10, 35, 50, 100, 150, 200\}$.

For each random forest, the misclassification rate was calculated using each of two methods:

(i)  the OOB sample; and
(ii)  the misclassification rate of the observations in the testing data.

In addition, for each random forest, the variable importance was calculated using each of five methods:

(i)  the original MDI method, *sensu Breiman (2002)*;
(ii)  the original MDA method, *sensu Breiman (2001)*;
(iii)  the Actual Impurity Reduction (AIR) importance (*Sandri & Zuccolotto, 2008*; *Nembrini, König & Wright, 2018*);
(iv)  the Holdout variable importance (*Janitza, Celik & Boulesteix, 2018*); and
(v)  the Independent Holdout method which is the Holdout method but using cross-validation folds which have been separated prior to encoding.

For each combination of parameters, 99 sets of data were generated and a random forest was trained with 500 trees and the Gini index splitting rule. The levels of the predictor variables were integer encoded according to the alphabetical ordering of the levels and the average misclassification rate and VIM were recorded for each method. The process was then repeated with the levels of the predictor variables being target encoded based on class probabilities.

The `ranger()` function from the R package `ranger` (*Wright & Ziegler, 2017*) offers both target-based and target-agnostic encoding options internal to the function and was used for the analysis; however, analysis by a different implementation using pre-encoded predictor variables would lead to equivalent results.

### Code availability

All analyses were carried out using R version 4.3.1 (*R Core Team, 2023*) and the `ranger` package ("RANdom forest GEneRator") version 0.15.1 (*Wright & Ziegler, 2017*). The R code used in this study is available at https://github.com/smithhelen/OutOfTheBag/releases/tag/v.1.0.0. This includes the code to generate the simulated data for reproducibility.

## RESULTS

### Out-of-bag (OOB) error

In the ideal case of balanced data with random assignment of individuals, the misclassification rate with simulated data was expected to be $\frac{2}{3} \approx 0.67$ regardless of the sample size, number of predictor levels, or method of encoding predictor variables. This was indeed the case when the misclassification rate was calculated for a fully withheld

independent test set—except with a small sample size of 20. However, the internally calculated OOB error rate depended on the method used to encode the levels of the categorical predictor variables. When predictor levels were integer encoded based on alphabetical placement, the misclassification rate was 0.67, as expected; however, when the predictor levels were target encoded based on the first principal component of the weighted covariance matrix of class probabilities, the misclassification rate decreased with increasing numbers of factor levels, and this was compounded with smaller sample sizes (Fig. 5). The bias was further exacerbated with increasing number of predictor variables (Fig. S1).

### Variable importance measures

The average variable importance was also expected to be impervious to the method of encoding of predictor variables, and, under random assignment of variable levels and of individuals, the variable importance was expected to be zero. The independent-holdout method was the only method that returned the expected outcome (_i.e._ zero importance for both target-agnostic and target-based encoding methods). The MDI measure, which is calculated on in-bag samples, was not affected by the choice of encoding; however, MDI increased with both sample size and number of variables for both target-agnostic and target-based encoding methods. Each of the other three variable importance measures were influenced by the choice of encoding method. Although the Holdout method does not directly use OOB samples for its calculations, because it is performing the predictor encoding on the entire dataset, prior to splitting into cross-validation folds, it is affected in the same manner. When predictor levels were integer encoded (_i.e._, target-agnostic), the variable importance values were zero as expected; however, when the predictor levels were target encoded, the average variable importance increased with increasing numbers of factor levels. For the MDA and Holdout methods, this was compounded with smaller sample sizes, but the opposite was true for AIR, which showed greater bias for larger sample sizes (Fig. 6). In contrast with the OOB misclassification rate, the positive bias diminished with increasing number of predictor variables (Fig. S2).

## DISCUSSION

Random forest predictive models are well suited to data sets containing a large number of categorical predictors and/or predictors containing many levels. Such data presents challenges for predictive models including absent levels (_i.e._ levels of a predictor variable that are present in data for prediction that were not present when the random forest was trained) and high computational demands. In these cases, one-hot encoding is not recommended as it frequently leads to a prohibitively large number of binary variables. Ordinal encoding, however, may improve both predictive performance and efficiency of models and offers a solution to the 'absent-levels' problem (_Smith et al., 2024_).

Methods of ordinal encoding of categorical predictors may be dependent or independent of the target variable. Target-based methods of encoding, including the two-class optimisation employed by `randomForest` and `ranger`, and ordering according to the first principal component of the weighted matrix of class probabilities, as implemented

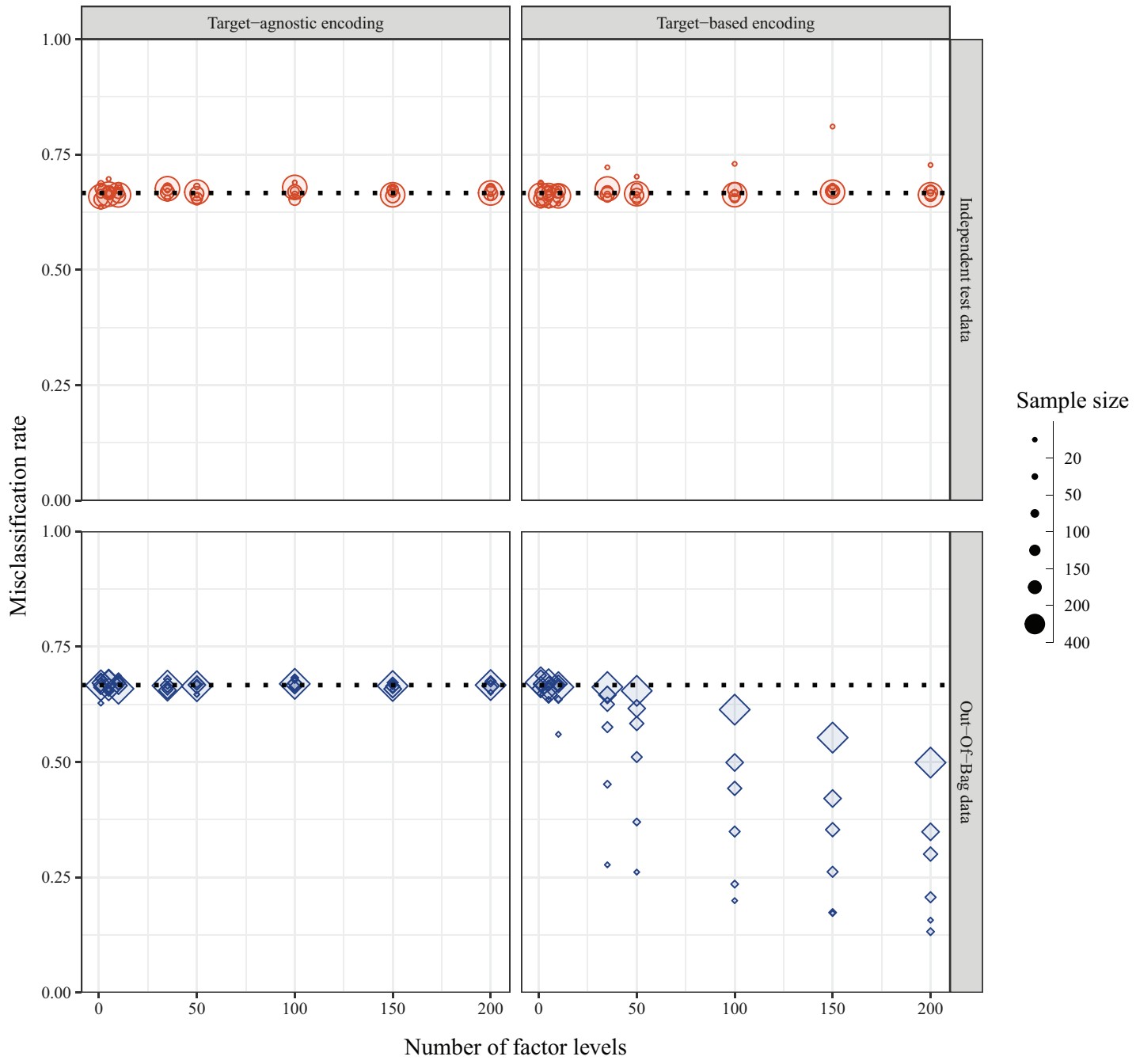

**Figure 5 Misclassification rates of data simulated with balanced design and random assignment of individuals to one of three classes as calculated *via* independent test set (top panel, red circles) and internal OOB sample (bottom panel, blue diamonds) when the method of encoding predictor variables is target-agnostic (ordered (alpha)numerically, left panel) or target-based (ordered *via* principal component analysis (PCA) of class probabilities, right panel).** The dotted line indicates the expected misclassification rate under the simulated null conditions.

[10] Also the CA-unbiased variation described in *Smith et al. (2024)*.

in `ranger`[10], use information from the target variable to inform the ordering. When the encoding is performed prior to bagging, there is leakage of information from the target variable to the observations in the OOB set. The leakage occurs because, even when the

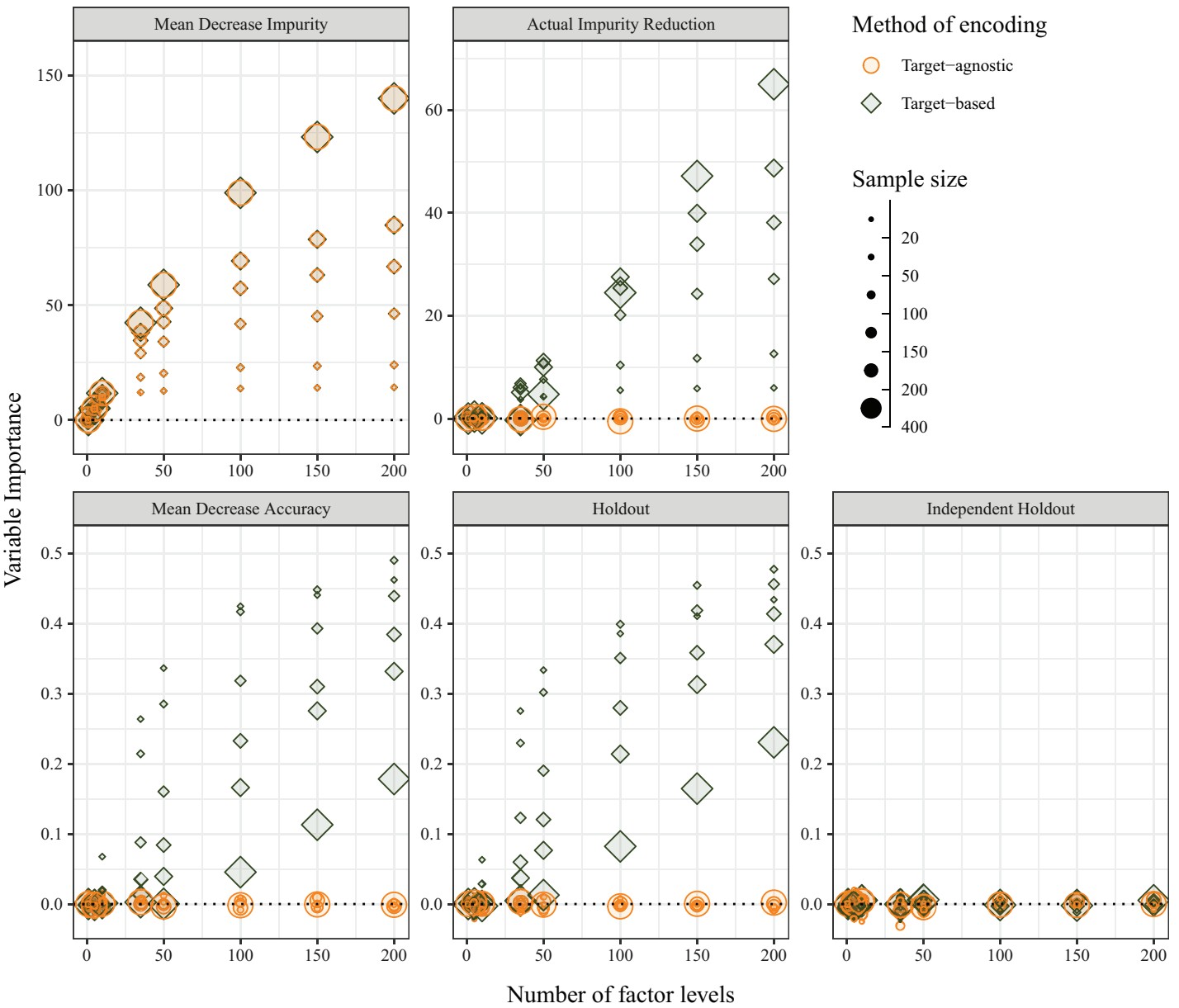

**Figure 6** Average variable importance as calculated using the five methods when the method of encoding predictor variables is target-agnostic (circles; encoded as integers) or target-based (diamonds; encoded *via* principal component analysis (PCA) of class probabilities).

observations are out of bag, the encoding of their corresponding levels was informed from the entire training dataset (*i.e.*, prior to the observations moving OOB) based on the correct response classes (*i.e.*, the target). This means the OOB observations do not behave like fully independent test data.

Target-agnostic methods of encoding, such as the naïve alphabetical encoding, or ordering according to some characteristic of the data (*e.g.* the PCO-encoding method (*Smith et al., 2024*)), are not subject to the issue of data leakage because the levels are

encoded using data on the predictor variables only–the response class (target) information is not used. Therefore, in these cases, it is entirely appropriate to treat OOB observations like fully independent test data.

*Breiman (1996, 2001)* claimed that the out-of-bag sample was as reliable as using an independent set of data for testing. We found that, for random forests, different methods of encoding nominal variables had important implications for the accuracy of calculations performed on out-of-bag samples. We showed that the OOB misclassification rate, and the variable importance measures which utilise OOB samples (the MDA, Holdout, and AIR measures), were biased when using a target-based encoding method due to 'data-leakage' during the *a priori* encoding of categorical predictors. When the encoding method is target-based, and the encoding is performed prior to bagging, the OOB data underestimates the true rate of misclassification, and overestimates true variable importance.

In all cases the bias increases with increasing number of factor levels, and is influenced by sample size. The effect of information leakage on OOB misclassification rates is more pronounced with smaller sample sizes and leads to lower misclassification rates (higher accuracy). The information leakage does not affect the permuted variable, as the relationship with the target is broken, and therefore MDA and Holdout variable importance measures both increase with decreasing sample size. In contrast, when variable importance is measured using purity of splits (*e.g.* the AIR method), rather than misclassification rates, information leakage has a more pronounced effect when sample sizes are larger. Although the MDI measure is not affected by method of encoding, it is also dependent on sample size. For both the MDI and the AIR methods, increasing sample size results in better purity of splits leading to higher variable importance values. MDI is known to be biased in favour of variables with many possible split points (*Strobl et al., 2007*). Larger sample sizes represent a greater number of factor levels, and therefore split points, which is artificially inflating variable importance.

A potential solution to the problem of information leakage to the OOB sample is to order the levels of each bootstrap sample independently (*i.e.* rather than ordering once on the entire dataset prior to bagging) (Fig. 4). We note that there are currently no implementations of random forest which offer encoding after bagging for the multiclass case. Another option is to calculate misclassification rates and variable importance measures on truly independent test data.

The findings of our study have several important research and practical implications for machine learning practitioners. Our aim is not to recommend a particular VIM or error estimation technique, but rather to discard the belief that OOB samples are a replacement for independent test data in all instances. This is not an issue for numeric or ordinal data. But for nominal categorical predictors which are ordinal encoded using a target-based encoding method, we recommend calculating misclassification rates from a separate, fully independent, test dataset; and calculating variable importance *via* MDA using an independent test set as the holdout sample.

## CONCLUSION

This article highlights how different methods of encoding of categorical predictors can bias OOB misclassification rates and variable importance measures. For datasets with a high number of variables and/or variable levels, absent levels are likely and ordinal encoding is a sensible approach for both speed of analysis and accuracy of predictions. When levels of categorical predictor variables are target encoded using class probability information and when encoding occurs prior to bagging, OOB samples suffer information leakage and are not a replacement for an independent test set. Using OOB data in place of an independent test set will lead to inflated measures of accuracy and variable importance. These findings are applicable to random forests and other tree-based methods (*e.g.*, boosted trees) where OOB misclassification rates and/or variable importance measures are calculated.

### Funding

This research is supported by a Massey University School of Fundamental Sciences scholarship. The New Zealand Food Safety Science & Research Centre (NZFSSRC) has paid for the APC of this article. The funders had no role in study design, data collection and analysis, decision to publish, or preparation of the manuscript.

### Grant Disclosures

The following grant information was disclosed by the authors:
Massey University School of Fundamental Sciences scholarship.
New Zealand Food Safety Science & Research Centre (NZFSSRC).

### Competing Interests

The authors declare that they have no competing interests.

### Author Contributions

- Helen L. Smith conceived and designed the experiments, performed the experiments, analyzed the data, performed the computation work, prepared figures and/or tables, authored or reviewed drafts of the article, and approved the final draft.
- Patrick J. Biggs conceived and designed the experiments, authored or reviewed drafts of the article, and approved the final draft.
- Nigel P. French conceived and designed the experiments, authored or reviewed drafts of the article, and approved the final draft.
- Adam N. H. Smith conceived and designed the experiments, authored or reviewed drafts of the article, and approved the final draft.
- Jonathan C. Marshall conceived and designed the experiments, performed the experiments, analyzed the data, performed the computation work, authored or reviewed drafts of the article, and approved the final draft.

### Data Availability

The code used for the random noise simulation study is available in the Supplemental File, at GitHub and Zenodo:

- https://github.com/smithhelen/OutOfTheBag/releases/tag/v.1.0.0.
- HLSmith. (2024). smithhelen/OutOfTheBag: Out Of The Bag PeerJ submission (v.1.0.0). Zenodo. https://doi.org/10.5281/zenodo.13755849.

The code generates data, encoded by both a target-based and a target-agnostic method, for analysis by random forest and calculates OOB error and VIM as described in the methods.

## Supplemental Information

Supplemental information for this article can be found online at http://dx.doi.org/10.7717/peerj-cs.2445#supplemental-information.

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
