# Peer review of "Out of (the) bag—encoding categorical predictors impacts out-of-bag samples"

_PeerJ Computer Science, doi:10.7717/peerj-cs.2445_

## Round 0.1 · original submission · Major Revisions

Please find the attached comments for your reference. Please fully address all the comments and submit the revised version of the manuscript for further considerations.

Reviewer 1 ·

Basic reporting

This study examines how target-based versus target-agnostic encoding of categorical predictor variables in random forests can bias performance metrics based on OOB samples. The authors demonstrate that when categorical variables are encoded using a target-based method and the encoding is done before bagging, the OOB sample can underestimate the actual misclassification rate and overestimate variable importance. Therefore, They recommend using a separate test data set to evaluate variable importance and the predictive performance of tree-based methods that employ a target-based encoding approach. The paper has an objective and can applied in many applications. These are the comments for this work to improve it more clearly.

1. In the introduction, authors should discuss the broader field, including current debates or issues relevant to their research. This helps to provide readers with the necessary background to understand the significance of the study. The introduction should also clearly state the problem, knowledge gap, or specific question the research aims to address. This way, readers can see how the research fits into the larger context and why it is essential.
2. Your paper is missing a literature review. Including relevant studies demonstrates that you are familiar with the key discussions happening in your field, particularly about out-of-bag (OOB) performance measures. By citing specific works, you can identify gaps in existing research, justifying the need for your study on target-based versus target-agnostic encoding of categorical predictor variables for random forests. Reference studies that support your investigation into bias performance measures based on OOB, challenge current assumptions, or provide a theoretical framework for your work. This shows that your research is not isolated but actively engages with ongoing issues or debates within your discipline.
3. The research process requires more detailed explanations. If possible, provide illustrations to enhance understanding, as well as detailed information about the data used in the experiments.
4. Figure 4 assumes that the encoding method and the evaluation metric are independent of each other. However, the interaction between encoding methods and OOB error calculations may vary depending on other factors, such as data distribution and model complexity. Please clarify this.
5. In this experiment, the author summarized the results using only the random forest method for prediction. To broaden the scope of the research, the study should compare the results with other state-of-the-art methods.

Experimental design

The experiment is complete within the context of what is required for this research.

Validity of the findings

The results of the experiment still have limitations. Comparative experiments with other tools are still needed in order to draw conclusions as required by the research.

Additional comments

-

Cite this review as

Reviewer 2 ·

Basic reporting

The paper contains an implemetation of OOB method in random forest.
The paper dosn't gives any novelty (nor proposes a new model)
No keywords
The introductions is too long and it is in the form of sub-sections
The mothodlogy is not presented

Experimental design

Author implemts and simulate exsted method

Validity of the findings

poor

Additional comments

The paper can not be accepted in the current form

Cite this review as

Reviewer 3 ·

Basic reporting

The paper investigates how target-based versus target-agnostic encoding of categorical predictor variables for random forest can bias performance measures based on out-of-bag (OOB) samples. The authors argue that when categorical variables are encoded using a target-based encoding method and when the encoding takes place before bagging, the OOB sample can underestimate the valid misclassification rate and overestimate variable importance. The Introduction section presents the concepts of Out-of-bag samples and out-of-bag error. It also discusses the variable importance measures (VIM) and issues related to encoding categorical predictors.

The Introduction section does not present the goal of the study nor the research question to be investigated. It discusses the relevant topics for the study but needs to establish their relationship and importance for the research goal. Another limitation of the Introduction section is the need for a description and presentation of the structure of the paper. Those aspects improve the readability of the study.

The text does not inform the availability of raw data for validation and replication purposes. For example, there is no clear indication of the availability of the simulated data despite its description in line 170: “The simulated data consisted of n individuals, each with one predictor variable allocated uniformly and with replacement from k levels.” The code availability note in line 200 indicates a repository that does not contain the simulated data. If the simulated data is created by the “out_of_the_bag_code.R” script available in the URL GitHub repository informed in line 203, it should be appropriately informed in the text.

Experimental design

As mentioned before, the aims and scope of the study should be clearly stated, including the presentation of a research question.

In line 167 in the Methods section, the text informs that data was simulated and analyzed with random forest to investigate the accuracy of internally calculated misclassification rates and variable importance under null conditions. This should have been described in the introduction section, along with a concise description of the results.

Validity of the findings

The text needs to discuss the validity of the findings. I strongly recommend to include internal, external, construction, and conclusion validity.

Additional comments

Table S1 “Implementation specific treatment of categorical variables” is essential for the discussion from line 121 to line 164. For this reason, the text could have used its structure to discuss and compare specificities in treating categorical variables.

Cite this review as

---

## Round 0.2 · accepted · Accept

Congratulations, the reviewers are satisfied with the revised version of the manuscript and have recommended the acceptance decision. While in production, please address all the minor formatting and grammatical issues.

Reviewer 1 ·

Basic reporting

In this version of the work, the author has made adjustments to a good standard, with revisions made as specified.

Experimental design

The experimental design has been clearly specified.

Validity of the findings

The findings indicate that when categorical variables are encoded using a target-based encoding technique, and this encoding is performed before bagging, the out-of-bag (OOB) sample may underestimate the actual misclassification rate and overestimate the importance of variables.

The order of operations in machine learning preprocessing can significantly affect the evaluation metrics. Specifically, it reveals that performing target-based encoding of categorical variables before applying bagging can lead to misleading evaluation results. The out-of-bag (OOB) sample, which is often used to estimate the performance of a model, may give inaccurate estimates of misclassification rates and overstate the importance of certain variables.

Additional comments

I am satisfied with the work in this version. There are only minor revisions for the author to address, such as language structure and various formatting issues.

Cite this review as

Reviewer 3 ·

Basic reporting

No comment.

Experimental design

No comment.

Validity of the findings

No comment.

Additional comments

The reviewed text is much better and improved its clarity and objectivity to address why the common practice of using OOB samples instead of independent test data can lead to biased and potentially misleading results due to information leakage from the target variable while encoding categorical predictors.

Cite this review as